# Sphingosine-1-Phosphate (S1P) and S1P Signaling Pathway Modulators, from Current Insights to Future Perspectives

**DOI:** 10.3390/cells11132058

**Published:** 2022-06-29

**Authors:** Gary Álvarez Bravo, René Robles Cedeño, Marc Puig Casadevall, Lluís Ramió-Torrentà

**Affiliations:** 1Girona Neuroimmunology and Multiple Sclerosis Unit, Department of Neurology, Dr. Josep Trueta University Hospital, Girona Biomedical Research Institute (IDIBGI), 17007 Girona, Spain; garyalvarez.girona.ics@gencat.cat (G.Á.B.); rrobles.girona.ics@gencat.cat (R.R.C.); mpuigcasadevall.girona.ics@gencat.cat (M.P.C.); 2Medical Sciences Department, Faculty of Medicine, University of Girona, 17004 Girona, Spain; 3Instituto de Salud Carlos III, Redes de Investigación Cooperativa Orientada a Resultados en Salud (RICORS), Red de Enfermedades inflamatorias (RD21/0002/0063), 28220 Madrid, Spain

**Keywords:** S1P, S1PR1, fingolimod, ozanimod, siponimod, multiple sclerosis

## Abstract

Sphingosine-1-phosphate (S1P) and S1P receptors (S1PR) are bioactive lipid molecules that are ubiquitously expressed in the human body and play an important role in the immune system. S1P-S1PR signaling has been well characterized in immune trafficking and activation in both innate and adaptive immune systems. Despite this knowledge, the full scope in the pathogenesis of autoimmune disorders is not well characterized yet. From the discovery of fingolimod, the first S1P modulator, until siponimod, the new molecule recently approved for the treatment of secondary progressive multiple sclerosis (SPMS), there has been a great advance in understanding the S1P functions and their involvement in immune diseases, including multiple sclerosis (MS). Modulation on S1P is an interesting target for the treatment of various autoimmune disorders. Improved understanding of the mechanism of action of fingolimod has allowed the development of the more selective second-generation S1PR modulators. Subtype 1 of the S1PR (S1PR1) is expressed on the cell surface of lymphocytes, which are known to play a major role in MS pathogenesis. The understanding of S1PR1’s role facilitated the development of pharmacological strategies directed to this target, and theoretically reduced the safety concerns derived from the use of fingolimod. A great advance in the MS treatment was achieved in March 2019 when the Food and Drug Association (FDA) approved Siponimod, for both active secondary progressive MS and relapsing–remitting MS. Siponimod became the first oral disease modifying therapy (DMT) specifically approved for active forms of secondary progressive MS. Additionally, for the treatment of relapsing forms of MS, ozanimod was approved by FDA in March 2020. Currently, there are ongoing trials focused on other new-generation S1PR1 modulators. This review approaches the fundamental aspects of the sphingosine phosphate modulators and their main similarities and differences.

## 1. Introduction

Sphingosine-1-phosphate (S1P) and S1P receptors (S1PR) are bioactive lipid molecules that are ubiquitously expressed in the human body and play a fundamental role in the trafficking and activation in the immune system. The first description by Spiegel et al. defined S1P as a signaling molecule that stimulates fibroblast proliferation [1,2]. S1P itself is a physiologic signaling molecule that acts as a ligand for a group of cell surface receptors. There are five different high-affinity cell surface receptors of S1P, which belong to the superfamily of G protein-coupled receptors (GPCR) [3]. Interaction with specific intracellular targets facilitates the propagation of S1P signals and subsequently, upon release, ligation of the five known heptameric GPCR can result both in autocrine and paracrine signaling [4].

The important role played by S1P/S1PR in various cellular functions such as vascular tone, heart rate, preservation of endothelial barrier and fundamentally, in immune cell trafficking have been proven in multiple studies [5]. This wide range of biological functions leads S1P/S1PR to be involved in pathogenesis of immunological and non-immunological conditions. Thus, S1P and S1PR were proposed as therapeutic targets for prevention or treatment of some diseases [6].

This review summarizes the fundamental aspects of the different sphingosine phosphate modulators, including their main similarities and differences, their application in the clinical practice, and their future perspectives.

### 1.1. Mechanism of Action of S1P/S1PR

The cell membrane is composed of a large number of proteins, among them sphingosines, which in turn are the source of many bioactive lipids, including ceramide, sphingosine, S1P and ceramide-1-phosphate [7,8]. S1P is the active terminal derivative of sphingosine metabolism, generated by the action of sphingosine kinase (SKs) [9,10]. SKs are activated by cytokines, immunoglobulins, hormones, and trophic factors [10]. S1P/S1PR complex has a broad range of functions due to the autocrine and paracrine effects and its presence in the blood and the lymphatic system (Figure 1).

S1P exerts its action through multiple enzymatic reactions in different cell locations, such as endoplasmic reticulum, mitochondria and nucleus [11,12].

The main S1P effect is regulating the lymphocyte egress from secondary lymphatic organs (SLO) into the systemic circulation, being the SLO the main location where its cardinal function is implemented [13,14,15]. To reach an adequate gradient of lymphocytes between blood and the lymph nodes a perfect equilibrium between the synthesis and degradation of S1P is necessary, via sphingosine kinases and sphingosine lyases, respectively. This equilibrium is altered when a modulator such as fingolimod is introduced [16,17].

S1P was initially thought not to have an active function, but this misconception changed when the S1PR was discovered and its characteristics as first messenger were better understood [18,19]. It has been proven that S1P is involved in multiple biological effects due to both its ubiquitous distribution and its pleiotropic effect, not only in the immune system, but also in limb development, neurogenesis, cardiogenesis, and generation and proliferation of vessels [20,21].

As was previously mentioned, S1P functions are mediated by five GPCR expressed in many organs (S1PR1 to S1PR5) [22]. S1PR1 expressed on lymphocytes T and B is implicated in modulating the expression of some pro-inflammatory cytokines [23,24,25]. S1P is involved in mechanisms of immune tolerance and prevention of autoimmunity. There are some ongoing trials about its implication on autoimmune diseases based on these effects [25,26].

S1P is released by cerebral sphingosines in the central nervous system (CNS), and its receptor is expressed by all types of brain cells, including neurons, astrocytes, and oligodendrocytes [27,28]. The discovery of the presence of the S1P/S1PR complex in the CNS was the main factor to figure out that the modulation of the signaling of this complex might have therapeutic implications for neurological disorders, including multiple sclerosis [29,30].

Given all the previously exposed, the effects of S1P can be theoretically divided into both peripheral effect due to the actions exerted on the immune system, and central effect because of its interactions in the central nervous system [31].

### 1.2. S1PR Isoforms

#### 1.2.1. Sphingosine-1-Phosphate Receptor: Isoform 1 (S1PR1)

S1PR1 is ubiquitous and is virtually expressed in every cell line [32]. Its wide distribution is the reason it has multiple biological effects in various organs [32,33]. As outlined above, S1PR-1 has a peripheral effect in the immune system and other organs and a central effect in the CNS [32]. Multiple functions such as actin remodeling, chemotaxis, lymphocyte egress, vascular integrity, organogenesis including angiogenesis, cell growth and proliferation and antimicrobial cytotoxicity are carried out by immune and vascular cells at peripheral level [33,34] (Figure 2). In CNS, S1PR1 is expressed by astrocytes, microglia, oligodendrocytes and neurons [33]. Its effects vary depending on the cellular lineage, not being well understood yet. Nevertheless, it is thought that S1PR1 induces overexpression of glial fibrillary acidic protein (GFAP) levels and morphological changes in neurons, therefore being directly involved in astrocyte proliferation and activation [35]. In neurons, S1PR1 also stimulates the migration of neuronal progenitor cells towards lesion sites [36]. The S1PR1 effects on oligodendrocyte progenitor cells (OPCs) are not properly stablished, but modulators such as fingolimod or siponimod have failed to show remyelination [34].

The fingolimod causes down-regulation in multiple lines of cells, so immunosuppressive effect is due to the downregulation of S1PR1 on T cells whereas its action on endothelial cells is responsible for increasing permeability and vascular leakage with the subsequent side effects observed after long-term treatment [36].

S1P1 seems to be a promising pharmacological target that remained to be explored in the different diseases in which it is pathophysiological involved.

#### 1.2.2. Sphingosine-1-Phosphate Receptor: Isoform 2 (S1PR2)

The S1PR2 binds to S1P with high affinity [37]. S1PR2, as well as other S1PR, signals through various G proteins and is widely expressed in different organs [38]. It is supposed that S1PR2 plays an important role in inhibition of apoptosis, cellular proliferation, actin remodeling and B cells positioning in follicles and in the the development of the heart and the auditory and vestibular system [39,40]. The establishing of endothelial barriers, especially in inner ear and retina are other of its key functions [39]. Nonetheless, these biological effects are poorly understood, thus the development of selective modulators has not been studied sufficiently [40,41].

#### 1.2.3. Sphingosine-1-Phosphate Receptor: Isoform 3 (S1PR3)

The functions of S1PR3 are still not completely clarified, but an essential role in the regulation of vascular tone by mean of vasodilation has been proposed. It is also involved in cytokine production, in the protection of myocardial ischemia, and in coagulation during inflammatory processes [42] (Figure 2). Moreover, S1PR3 enhances the Notch signaling pathway and is implicated in retinal astrogliosis. The S1PR3 effect on immune system remains controversial as both pro-inflammatory and anti-inflammatory effects have been shown [42,43]. Most of the research performed on S1PR3 is related to vascular contraction, stroke, sepsis, cardiac conductivity, asthma and cancer growth and metastasis formation [44,45].

The fingolimod effect on S1PR3 in the heart conduction system causes a negative chronotropic response [46].

#### 1.2.4. Sphingosine-1-Phosphate Receptor: Isoform 4 (S1PR4)

The S1PR4 is specifically expressed in SLO, hematopoietic tissue and lungs, where it plays an essential role in lymphocyte signaling, megakaryocyte differentiation and platelet formation and activation [47,48] (Figure 2). In CNS, S1PR4 mediates the activation and maturation of dendritic cells [47]. S1PR4 is a negative regulator of cells proliferation and participates in reducing the secretion of effector cytokines [48]. The functioning of S1PR4 system is currently under investigation, and some novel selective agonists are being studied to better understand its biological importance [49].

#### 1.2.5. Sphingosine-1-Phosphate Receptor: Isoform 5 (S1PR5)

Oligodendrocytes and myelinating cells of the brain and spleen are the cellular lines, which most widely express this S1PR isoform [50]. S1PR5 is one of the main regulators of natural killer cells egress from both bone marrow and spleen into the blood [51]. Activation of the S1PR5 on oligodendrocytes may have a beneficial effect in MS by protecting the oligodendrocytes from demyelination and cell death [52] (Figure 2). Immune quiescence and blood–brain-barrier integrity are other functions that could be mediated by the expression of S1PR5 in the brain endothelial cells [50,51].

## 2. S1P Modulators

The understanding of S1P functions, especially in the CNS, supposed a breakthrough for the developing of molecules with beneficial biological effect. From fingolimod until Siponimod, which has been the last one accepted, the S1P modulators constitute an interesting therapeutic option for treating some autoimmune disorders, including MS.

### 2.1. Fingolimod

Fingolimod received regulatory approval from the FDA in 2010 as the first-in-class S1PR modulator used for the treatment of the relapsing forms of multiple sclerosis (RMS) [53]. Fingolimod is a prodrug, which needs phosphorylation to exert its biological effect [54].

Fingolimod (FTY720) is a nonselective modulator of S1PR1, S1PR3, S1PR4, and S1PR5. At much higher concentrations, fingolimod can also activate other intracellular targets such as kinases and lyases of sphingosine, ceramide synthase, cytosolic phospholipase A, histone deacetyases, protein phosphatase A2, and the cation channel TRPM7 [54].

Fingolimod impedes the lymphocyte egress from lymph node due to its antagonism of S1PR1, causing a functional sequestration of B cells and naive and central memory T cells and subsequently reducing the infiltration of inflammatory cells into CNS. Fingolimod does not interfere with lymphocyte activation [13,55,56,57]. The effect on circulating lymphocytes is dose-dependent, decreasing by 20% to 30% within the first week of treatment [58,59] (Figure 2 and Figure 3).

The FREEDOMS trial showed fingolimod effectiveness compared to placebo for decreasing the annualized relapse rate (ARR) in patients with multiple sclerosis, reaching the clinical and radiological proposed endpoints. Fingolimod also demonstrated a significant preservation of brain volume. FREEDOMS II confirmed these outcomes providing strong clinical evidence for its use [60].

The TRANSFORMS trial, which compared Fingolimod with intramuscular interferon (IFN)-β1demonstrated that the reduction of ARR was greater in both fingolimod groups (two fingolimod doses were used). MRI endpoints were also reached. Nevertheless, there were no significant differences in time to confirmed disability progression [61,62]. The lack of fingolimod effectiveness in modifying the disability progression was corroborated in MS (INFORMS) trial [63].

The benefit/safety profile of fingolimod have been well stablished, although its use can cause complications such as first-dose bradycardia and the occurrence of atrioventricular block, hypertension, pulmonary, liver toxicities, macular edema and reactivation of some latent viral infections. Most complications are likely mediated by the fingolimod interaction with other S1PR isoforms [64,65].

The fingolimod effects are not exclusively S1P-dependent, and due to its structural similarity to sphingosine, fingolimod can inhibit cancer-relevant signal transduction independently of S1PR binding. This property facilitates its involvement in different functions related to cancer pathophysiology such as angiogenesis, cell migration, proliferation and invasion, metastasis and inflammation. Notwithstanding, further investigations should be warranted to repurpose fingolimod as a novel anti-cancer therapy [66,67].

### 2.2. Siponimod

Siponimod is a novel alkoxyimino derivative and a functional antagonist of S1PR1 and S1PR5. S1PR5 are expressed by neural cells such as astrocytes, oligodendrocytes, microglia or neurons [11,12]. Like fingolimod, siponimod induces lymphopenia by sequestration in lymph nodes. [68] The main difference of siponimod compared to fingolimod is the capacity to cross the blood–brain barrier and to exert its functions directly on the brain cells (Figure 4) impeding synaptic neurodegeneration according conclusions of preclinical studies [69]. Gentile et al. delivered siponimod directly into the brain by means of continuous intracerebroventricular infusion to evaluate its effects on CNS cells. This experiment showed that fingolimod prevents oligodendrocyte degeneration and, thus, ameliorates induced-demyelination in experimental autoimmune encephalomyelitis (EAE) models. Moreover, neuronal loss and glial reactions were less severe in siponimod-treated mice [69,70].

Siponimod tropism over S1PR1-S1PR5 serves to reduce cardiac-related side effects, such as bradycardia, observed in patients treated with fingolimod [71].

Siponimod effectiveness for RRMS and SPMS was demonstrated by BOLD and EXPAND studies, respectively [71,72,73].

EXPAND trail met its both primary and secondary endpoints showing a significant reduction in both 3-month confirmed disability progression and T2 lesion load, GdE lesions, and new or enlarging T2 lesions, respectively. Moreover, loss of brain volume was lower in the Siponimod group. Siponimod is a potent anti-inflammatory drug—its effect was greater in a subgroup of patients with inflammatory activity according the EXPAND study—but due to its mechanism of action, at the same time attenuates the neurodegeneration [73]. The outcomes obtained from EXPAND trial supposed a breakthrough in the SPMS treatment, as there were no approved options to treat this form of MS.

Siponimod reduces lymphocyte counts within 4 to 6 h, but it also has a much shorter half-life than fingolimod, which facilitates to recover lymphocyte counts to baseline levels within a week after treatment discontinuation [74].

Summarizing, siponimod effects can be explained by four main mechanisms: (i) Modulation of the reactivity of glial cells by ameliorating neuronal and/or oligodendrocyte injury. (ii) Stabilization of the blood–brain barrier due to modulation of microglia and astrocytes. (iii) Immune cell recruitment. (iv) Absence of peripheral immune cells into the CNS [70].

### 2.3. Ozanimod

Ozanimod is a potent agonist of the S1PR1 and S1PR5. Unlike fingolimod, it does not need phosphorylation to be an active molecule and as well as siponimod the lack of interaction with S1PR3 minimizes potential safety concerns [75,76].

In models of EAE ozanimod showed attenuation of immune responses by reduction of circulating lymphocytes [76]. Ozanimod targets some specific T lymphocyte subsets implicated in the pathophysiology of autoimmune disorders such as MS without interfering on protective immunity. This property is achieved because of its effect was greater on central memory cells than effector memory cells. Additionally, ozanimod caused those levels of the lymphocyte subsets expressing cytokine receptor 7 (CCR7) were lower than other subsets [76].

Ozanimod induces rapid and dose-dependent reductions in absolute lymphocyte count, but also a rapid lymphocyte recovery after treatment discontinuation because of its short half-life (19–20 h) [77].

Ozanimod demonstrated to reduce plasma levels of neurofilament light chain. From this finding, neuroprotective properties were likely proposed [78].

RADIANCE study was decisive for approval of ozanimod by FDA. The trial met its endpoint—reduction of cumulative load of total GdE MRI lesions at weeks 12 to 24—compared with placebo [79].

In general terms, ozanimod showed a favorable safety profile with no notable cardiovascular, pulmonary, ophthalmic, infectious, or malignant events; nevertheless, evaluations of respiratory function are clinically indicated during its use, as it could be associated with deleterious respiratory effects in some patients [80].

Ozanimod is currently undergoing phase III trials for the treatment of both active ulcerative colitis and Crohn’s disease in various countries worldwide, as well as trials for the evaluation of cognitive processing speed changes in MS patients [81,82].

### 2.4. Ponesimod and Others S1P Modulators

#### 2.4.1. Ponesimod

Ponesimod is an orally active S1PR1 selective and rapidly reversible functional antagonist, with potential immunomodulating activity [83,84,85]. The receptor is transiently activated and then internalized, which leads to its degradation and, as a result, the sequestration of lymphocytes in the lymph nodes. This prevents a lymphocyte-mediated immune response, by reducing the amount of circulating peripheral lymphocytes and its infiltration into target tissues [84]. The effect of ponesimod differs depending on the lymphocyte subpopulations, causing a reduction in the count of T cells (CD3+) and B cells (CD20+) in a dose-dependent pattern, while no measurable modification on natural killer cells (CD16+). The return to a normal count for both B and T cells was reported within 48 h after a single dose [86]. Ponesimod has a shorter half-life and and faster elimination (within 1 week) compared to fingolimod which makes its pharmacological rapidly reversible [86].

Ponesimod can be useful to treat several inflammatory conditions as it has proven to modulate some responses of the inflammatory cascade such as inflammatory cell infiltration, edema formation, and proinflammatory cytokine levels [87].

The OPTIMUM study showed the superiority of ponesimod compared to teriflunomide in reducing the annualized relapse rate, the Fatigue Symptom and Impact Questionnaire–Relapsing Multiple Sclerosis symptom score, and the active lesions on magnetic resonance imaging in RRMS patients [88]. Ponesimod, compared to other S1P modulators, provides higher safety, but potentially earlier return of the inflammatory activity.

#### 2.4.2. Others S1P Modulators

S1P modulators such as ceralifimod (selective S1PR1 and S1PR5 modulator), GSK2018682 (selective S1PR-1 and S1PR-5 modulator), and amiselimod (MT-1303) (selective S1PR-1 modulator) have been developed by different drugs companies to treat patients with multiple sclerosis. However, despite their early outcomes, all of them were prematurely suspended due to private matters of sponsors [89,90,91].

## 3. Similarities and Differences

### 3.1. Similarities

The S1PR axis has been implicated in several immune-mediated disorders due to its pleiotropic functions, ubiquity and expression in different organs. All S1PR modulators link to S1PR1, and therefore all of them alter the gradient of lymphocytes between peripheral lymph organs and blood. By this pathway, their main biological effects are obtained, with all their desired therapeutic implications [60,80,88,90]. MS is the disease where the S1P mechanism has been most widely studied, but there are ongoing trials to evaluate a similar beneficial effect on other pathological conditions such as rheumatoid arthritis, inflammatory bowel diseases and others some types of cancers.

Fingolimod is the universal S1PR modulators due to its effect on all the isoforms, except S1PR2 [60]. Siponimod and ozanimod selectively modulate both S1PR1 and S1PR5, and most of their effects derive from this linkage. Ozanimod and Siponimod have demonstrated potentially neuroprotective effects on the CNS by the S1PR5 mediation [73,79].

No S1PR modulator is currently approved for use during either pregnancy or breastfeeding. In case of an exposed pregnancy, the treatment must be discontinued and the pregnancy closely monitored [63,71,77,81].

### 3.2. Differences

Fingolimod needs to be phosphorylated by SK2 to form the active metabolite fingolimod-(S)-phosphate. It has a wide spectrum of effects due to its pharmacodynamic properties on S1PR1, S1PR3, S1PR4 and S1PR5 [70]. This fact also implies that fingolimod has more side effects than other S1PR1 modulators. The agonistic effect on S1PR3, S1PR4 and S1PR5 could be the cause for some of the adverse effects observed, as mentioned before: bradycardia, hypertension, macular edema, reduced pulmonary function and neoplasms, which are not as frequently observed in other S1PR modulators [65].

Fingolimod has been suggested to be associated with a borderline but significant increased risk of cancer development [92]. However, it has also been suggested that by antagonizing S1PR3 in the presence of high levels of S1P, it could have anti-metastatic activity [93]. Further research is needed due to this apparent contradiction, and there are ongoing studies to try to elucidate the association between fingolimod and cancer.

Unlike fingolimod, others S1PR modulators do not need phosphorylation for activation. Siponimod and ozanimod act on isoforms 1 and 5, which bestow a neurotropism that favors effects on myelination by preventing oligodendrocyte degeneration [70]. They have both been tested to treat RRMS and SPMS [73,78].

Siponimod is the only S1PR modulator that needs genotyping for CYP2C9, since a dose adjustment will be needed in patients who are intermediate metabolizers. Siponimod is contraindicated in poor metabolizers [94].

Besides the receptor selectivity, time for elimination (half-life: T^1/2^) and lymphocyte restoration after discontinuation are other differences among S1PR modulators (Table 1). Fingolimod T^1/2^ is 7 days while the T^1/2^ for Siponimod, Ozanimod and Ponesimod are 30 h, 19–20 h, and 33 h, respectively [95]. Lymphocyte restoration after discontinuation also varies among S1PR modulators, being 6 weeks, 1–10 days, 2–3 days, and 7 days for fingolimod, siponimod, ozanimod, and ponesimod, respectively [63,71,77,81,95].

The new S1PR modulators have a shorter half-life than fingolimod, which implies a shorter wash-out period. However, a shorter T^1/2^ is also associated with a faster lymphocyte restoration after discontinuation, which in turn would increase the risk of a hypothetic earlier return of the inflammatory activity [71,96,97].

## 4. Future Perspectives

S1P modulators are drugs responsible of multiple actions such as anti-inflammation, neuroprotection and neuromodulation. This wide spectrum facilitates to target simultaneously several of the different pathophysiological pathways implicated in MS.

Siponimod is the first drug to have been shown to be of benefit in SPMS. This fact “could open the door” for other drugs with a similar mechanism, but further research in well-identified disease phenotypes will be needed for these drugs to demonstrate clinical efficacy.

The novel compounds such as siponimod and ozanimod, due to their higher selectivity and shorter half-life, appear to have a higher efficacy and safety profile than fingolimod as DMT.

It may be worthwhile to develop global pharmacological approaches for S1PR modulators to treat other medical conditions such as rheumatoid arthritis, inflammatory bowel disease, systemic lupus erythematosus, polymyositis, dermatomyositis and other neuromuscular disorders. In addition, due to their interference with angiogenesis, research on their potential as cancer treatments has also been made. However, as higher doses of S1PR modulators are required to obtain the desired effect, it has been assumed that none of them will achieve clinical responses at tolerable doses.

S1PR modulators could be potentially used in conditions with secondary autoimmune responses, as cerebrovascular diseases, due to its well-known inflammatory response during stroke [98]. Some studies tested the fingolimod’s usefulness in ischemic stroke demonstrating better outcomes compared to placebo in terms of circulating lymphocyte counts, neurological deficits, and better recovery [99]. Fingolimod treatment also reduced the neurologic impairment and perihematomal edema in patients with small-to-moderate sized deep primary supratentorial intracerebral hemorrhage [100]. Preclinical and clinical data suggested modulation of S1P/S1PR axis is a very promising target in stroke therapy, but further research is still warranted.

The scope of S1PR modulators is wide and new targets such as SKs, S1P lyase or S1P transporters that are currently being investigated will surely improve the modulation of the S1P axis.

## Figures and Tables

**Figure 1 cells-11-02058-f001:**
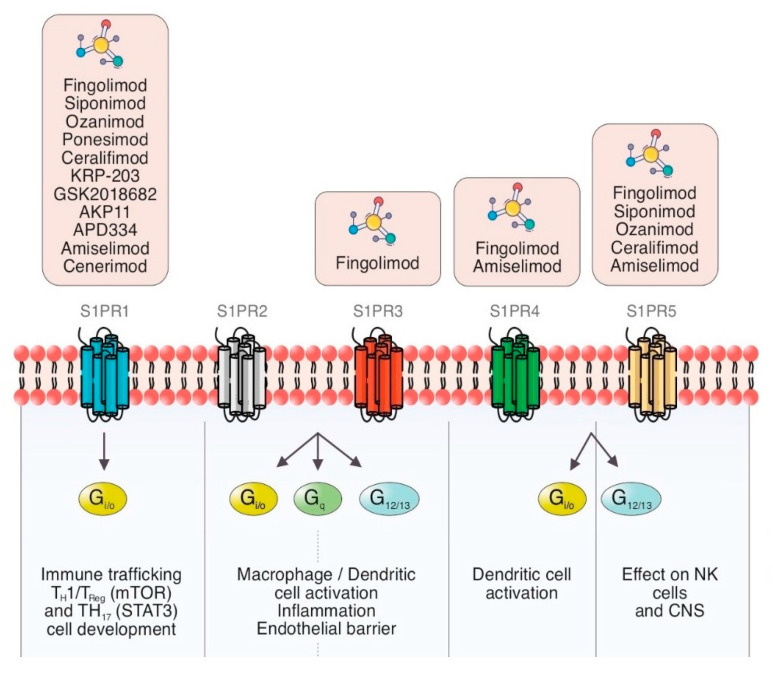
Schematic summary of the current view of S1PR modulators and S1P signaling pathways with cellular therapeutic targets in inflammation and immune processes through different G proteins.

**Figure 2 cells-11-02058-f002:**
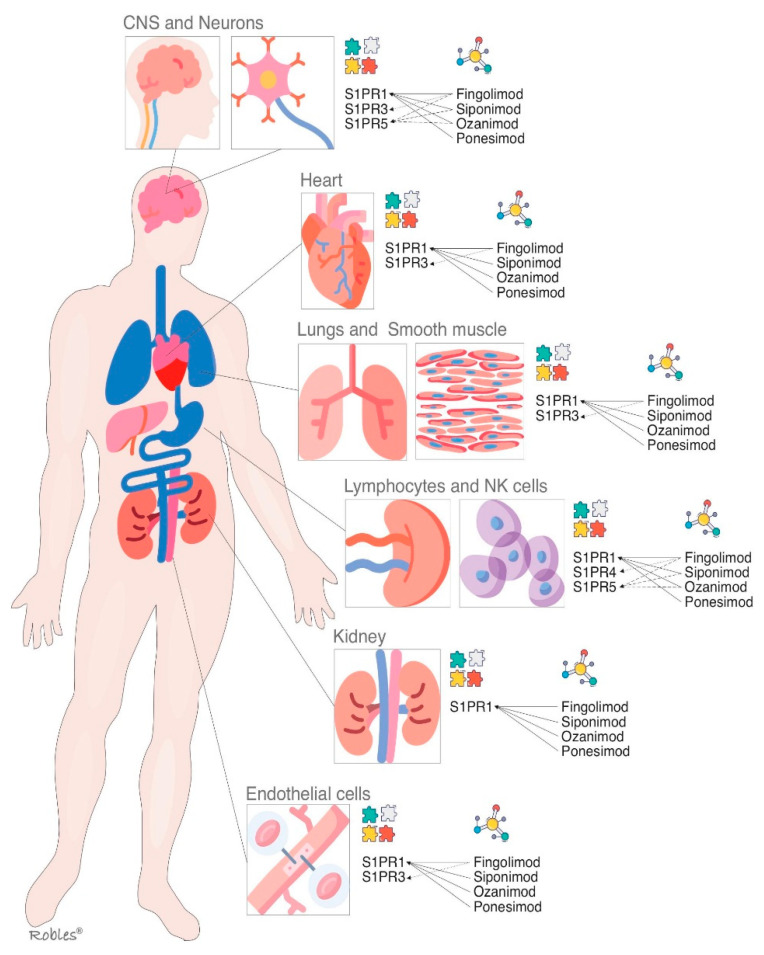
S1P receptors and their main localizations altogether with the S1P modulators and their targets. S1PR = Sphingosine1-phosphate receptor, CNS = central nervous system.

**Figure 3 cells-11-02058-f003:**
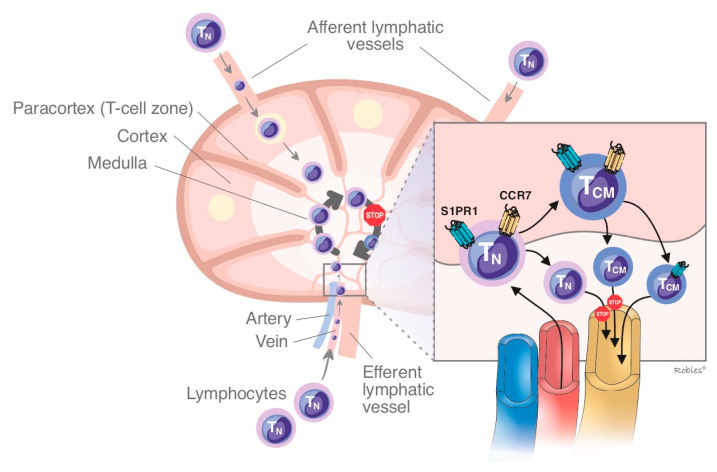
The effect of S1PR1 internalization in the lymph nodes and how regulates the gradient of lymphocytes between SLO/blood. This scheme shows the sequestration of naive T cells and CCR7 expressing central memory T cells due to the inability to response to the altered lymphocyte gradient between SLO/blood. S1PR = Sphingosine1-phosphate receptor; TN = naÏve T cells; TCM = central memory T cells; SLO = secondary lymphatic organs.

**Figure 4 cells-11-02058-f004:**
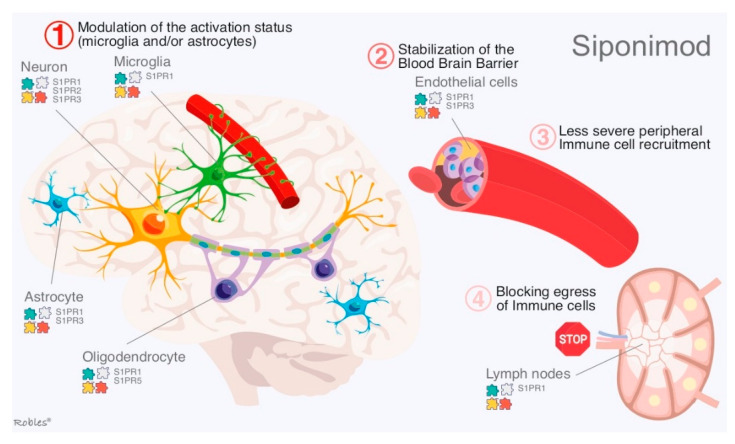
Mechanism of action of Siponimod: Part 1: interaction of Siponimod with all brain cells to produce an anti-inflammatory response and probable neuro-protective effect-Part 2: Siponimod effect on both the blood–brain barrier and the lymphocyte egress from lymph node to the blood.

**Table 1 cells-11-02058-t001:** Summary of different characteristics of the S1P modulators.

Drug	Receptor Selectivity	Genotyping Needed	Dosing	Elimination T^1/2^	Lymphocyte Restoration after Discontinuation
Fingolimod (Gilenya^®^) (FTY720)	S1PR1 S1PR3 S1PR4 S1PR5	No	0.5 mg/d	7 days	6 weeks
Siponimod (Mayzent^®^) (BAF312)	S1PR1 S1PR5	Yes	Depending on genotype: CYP2C9*1*1 and CYP2C9*1*2: 2mg/d CYP2C9*2*2 and CYP2C9*1*3: 1mg/d CYP2C9*2*3 and CYP2C9*3*3: contraindicated	30 h	1–10 days
Ozanimod (Zeposia^®^) (RPC1063)	S1PR1 S1PR5	No	0.92 mg/d	19–20 h	4–12 weeks
Ponesimod (Ponvory^®^) (ACT128800)	S1PR1	No	20 mg/d	33 h	7 days

## Data Availability

Not applicable.

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
