# Peer review of "Sphingosine-1-Phosphate (S1P) and S1P Signaling Pathway Modulators, from Current Insights to Future Perspectives"

_cells, 2022, doi:10.3390/cells11132058_

Round 1
Reviewer 1 Report
The authors have addressed my previous comments in this improved version of the manuscript. I am now recommending this study for publication.
Reviewer 2 Report
This is a well-structured and qualitative review about sphingosine-1-phosphate (S1P) and S1P signaling pathway modulators. The authors described in detail the biochemical and physiological features of the main molecules and noted promising areas for therapeutic use.
Strengths:
- The manuscript presents a balanced view of recent work by active groups in the subject area.
- The results of this study have valuable practical significance.
- The manuscript makes a valuable contribution to the field.
The paper deserves to be published in Your journal. No special corrections are required.
This manuscript is a resubmission of an earlier submission. The following is a list of the peer review reports and author responses from that submission.
Round 1
Reviewer 1 Report
The manuscript presented for evaluation concerns the issuesof cellular transmission taking place with the participation of S1P
and the receptors on which this agonist affects.
The work is clearly written, and the attached figures are
clear and very helpful in understanding the complex issues
of the effects of S1P and its synthetic analogues
on various receptors in different tissues.
The only missing element is a deeper analysis of the possibility
of using new S1P analogues in the treatment of cancers
and autoimmune diseases based on the literature data from the
last 3 years.
Reviewer 2 Report
Dear authors,
I have found a number of errors in the authors' interpretation of the references. In many cases, the references did not carry any information that the authors base their ideas on. Furthermore, the paragraphs are not at all organized and they contain too scattered information, thus it is very difficult to follow. I think the manuscript is not prepared yet for publication.
Reviewer 3 Report
The review by Bravo et al “Sphingosine-1-Phosphate (S1P) and S1P Signaling Pathway modulators. From current insights to future perspectives” is an attempt to summarize current knowledge about a few drugs that are able to modulate S1PR-mediated signaling and decrease inflammation in the context of neurological disorders. A significant effort is given to make this review, especially to provide nice colored figures. It is unfortunate, that behind this great work there is plenty of insufficiencies and incorrectness. Therefore, in the current form, this review cannot reach the audience as it sometimes misleads and provides a superficial view on the issue.
There are so many points that needs to be addressed. First, this is a review about FTY720, its analogs, and new ligands for S1PRs that got into clinical trials in MS and related disorders. Therefore, the review must have chemical structures of those drugs, with clear indication which receptors are targeted and in what form (this is described but in multiple places), provide information on drug/effect stability, therapeutic concentrations, etc. Next, a proper use of terminology is the must. We cannot say about S1P receptors that they are “modulators of S1P” or “bioactive lipid molecules”. “Sphingosine phosphate modulators” is also a term that has no meaning. If authors want to discuss intracellular direct S1P binding to its targets, this should be discussed, but currently, as it is written in Introduction, one could understand that S1P acts first inside the cells, interacts with its targets that results in its release, and only then S1P binds to its receptors.
Chapter 1.1. It is absolutely incorrect to say that sphingosine is the source of ceramides, S1P, ceramide-1-phosphate, and sphingosine itself. It is like there is no understanding about pathways of sphingolipid metabolism. Legend to Fire 1 states that S1PR2 and S1PR3 regulate endothelial barrier while S1PR1 is the most critical receptors that tightens it and the most expressed on endothelial cells.
In the entire review, there is no clear message that regardless pleiotropic expression of all five S1PRs, final outcome of S1P stimulation depends on their relative proportion on cell surface.
Page 4 – “S1P is released by cerebral sphingosines…” what is that nonsense?
Versions of receptor denomination are often used, for example, S1PR1, S1P1, and S1PR-1. Use only one version that is accepted – S1PR1 or S1P1,.
Page 6. “Heart condition system”. What is that? If you are talking about bradycardia then say that.
Term “S1P modulators”. This is an incorrect wording. You probably want to talk about modulators of S1P signaling? Or S1P analogs?
The authors almost completely ignore the seminal work of Tim Hla on lymphocyte egress and role of S1P and S1PRs in this process. At the end, authors do not explain the mechanism how exactly FTY720 works and blocks the egress of lymphocytes. Nothings is said about the work of Dr. Garcia and Dudek about S1PR1 ubiquitinilation and degradation by FTY720 (fingolimod). This message is diluted and not highlighted. Also, when talking about fingolimod, nothing is said about the resurgence of latent viral infections in people treated with the drug. Nothing is told about the inhibition of S1P lyase and ceramide synthases by FTY720.
Page 11. The sentence “This subtype selectivity suggests the possibility of ozanimod targeting T cells implicated in the immuno-pathology of conditions such as RMS whilst allowing protective immunity to be maintained” confuses as what was said before does not suggest at al what is stated. Detailed explanations are needed.
Wording like “different drug companies” should be replaced with exact names as you talk about specific molecules. And again, “S1P modulators” is incorrect, you are talking about S1PR ligands.
Page 14. Explanation is needed why you start talking about cytochromes. Does this mean that drug modifications by cytochromes is required for their action? Explain.